# DFlash: Block Diffusion for Flash Speculative Decoding

Jian Chen [1]   Yesheng Liang [1]   Zhijian Liu [1]

## Abstract

Autoregressive large language models (LLMs) deliver strong performance but require inherently sequential decoding, leading to high inference latency and poor GPU utilization. Speculative decoding mitigates this bottleneck by using a fast draft model whose outputs are verified in parallel by the target LLM. However, existing methods still rely on *autoregressive drafting*, which remains sequential and constrains practical speedups. Diffusion LLMs offer a promising alternative by enabling parallel generation, but current diffusion models typically underperform compared with autoregressive models. In this paper, we introduce **DFlash**, a speculative decoding framework that employs a lightweight block diffusion model for parallel drafting. We show that speculative decoding provides a natural and effective setting for diffusion models. By generating draft tokens in a single forward pass, DFlash enables efficient drafting, and by conditioning the draft model on context features extracted from the target model, it achieves high-quality drafts with higher acceptance rates. Experiments show that DFlash achieves over $6\times$ lossless acceleration across a range of models and tasks, delivering up to $2.5\times$ higher speedup than the state-of-the-art speculative decoding method EAGLE-3.

**Links:** Code (GitHub) | Models (Hugging Face)

## 1. Introduction

Large language models (LLMs) have enabled a wide range of powerful applications, including conversational agents (Yang et al., 2025; Guo et al., 2025) and automated programming tools. Despite their success, LLM inference remains dominated by a sequential, token-by-token generation process, where each output depends on the full preced-

ing context. This inherent seriality creates a major performance bottleneck: inference is slow, memory-bound, and fails to fully utilize modern GPUs. With the recent emergence of long Chain-of-Thought (CoT) reasoning models (OpenAI et al., 2024; Guo et al., 2025), this bottleneck has become increasingly critical, as prolonged inference times now dominate the generation process.

Speculative decoding (Leviathan et al., 2023; Li et al., 2025c; 2024; 2025b; Cai et al., 2024) has emerged as a primary solution to this bottleneck. This paradigm employs a lightweight *draft model* to speculate a sequence of future tokens, which are then verified in parallel by the large *target model*. While this approach achieves lossless acceleration and has been widely integrated into production frameworks, state-of-the-art methods like EAGLE-3 (Li et al., 2025b) still rely on autoregressive drafting. This serial drafting process is not only inherently inefficient but also susceptible to error accumulation, which effectively caps achievable speedups at approximately $2-3\times$.

Recently, Diffusion LLMs (dLLMs) (Nie et al., 2025) offer a promising alternative to autoregressive LLMs by enabling parallel text generation and bidirectional context modeling. Block diffusion models (Arriola et al., 2025; Cheng et al., 2025; Wu et al., 2025) can denoise a block of masked tokens simultaneously. However, current open-source dLLMs typically underperform their autoregressive counterparts in terms of generation quality. Furthermore, maintaining acceptable output quality often necessitates a high number of denoising steps, which significantly diminishes their raw inference speed (Qian et al., 2026).

This landscape reveals a critical trade-off: autoregressive models deliver superior performance but suffer from sequential latency, while diffusion models allow for fast, parallel generation but often at the cost of accuracy. A natural research question follows: *Can we combine the strengths of both paradigms while mitigating their respective weaknesses?* A compelling solution lies in leveraging diffusion models for high-speed, parallel drafting, while relying on high-quality autoregressive models for verification to ensure the final output remains lossless.

However, utilizing diffusion for drafting is non-trivial, and existing methods are either impractical or offer limited speedups. Methods such as DiffuSpec (Li et al., 2025a)

---

[1]UC San Diego. Correspondence to: Zhijian Liu <zhijian@ucsd.edu>.

*Proceedings of the 43rd International Conference on Machine Learning*, Seoul, South Korea. PMLR 306, 2026. Copyright 2026 by the author(s).

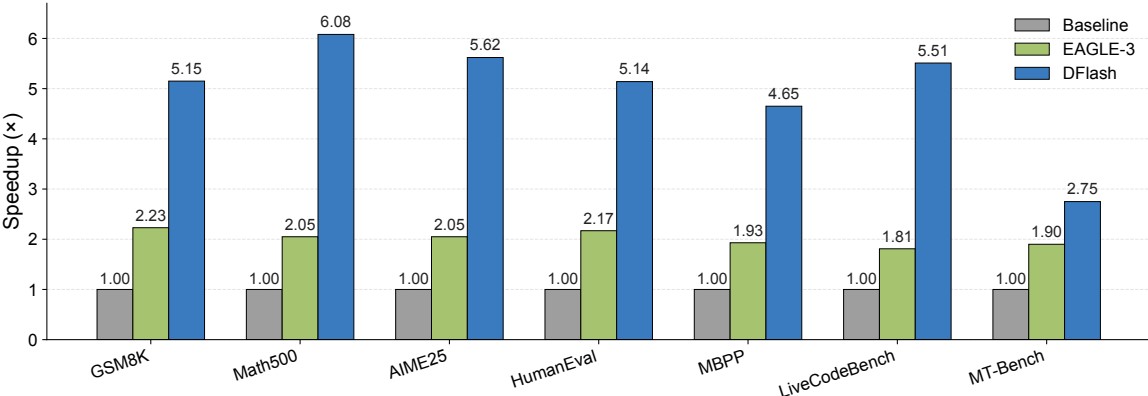

*Figure 1.* Speedup comparison between DFlash, EAGLE-3 against Autoregressive Decoding on Qwen3-8B (Yang et al., 2025) with the Transformers backend. Overall, DFlash achieves more than 2.5× higher speedup than EAGLE-3.

and SpecDiff-2 (Sandler et al., 2025) utilize massive (*e.g.*, 7B parameter) draft models. This significant memory footprint is often prohibitively expensive for real-world serving. Furthermore, while these large drafters offer relatively high quality draft tokens and acceptance lengths, the high drafting latency limits their practical speedups to a modest 3−4×. In contrast, PARD (An et al., 2025) trains small autoregressive models to mimic diffusion-style parallel generation, and then perform speculative decoding for target LLMs. However, the resulting small models lack the modeling capacity of the target LLMs, leading to limited acceptance lengths and a speedup ceiling of approximately 3×.

*Is there truly "no free lunch"? Can we build a diffusion drafter that is both lightweight and highly accurate?*

In this paper, we introduce DFlash, a speculative decoding framework that uses a lightweight block diffusion model to achieve both fast and high-quality drafting. Our key insight is simple: **the target knows best**. As observed by Samragh et al. (2025), large autoregressive LLMs' hidden features implicitly contain information about multiple future tokens. DFlash utilizes these hidden features as context, conditioning the draft model to predict future blocks of tokens in parallel. In effect, the draft model becomes a diffusion adapter that efficiently leverages the deep context features modeled by the large target model. Instead of requiring a tiny draft model to reason from scratch, DFlash fuses the reasoning capabilities of the target model with the parallel generation speed of a small diffusion drafter.

We evaluate DFlash across a wide range of models and benchmarks, and demonstrate its practical benefits under realistic serving setups using SGLang (Zheng et al., 2024). As shown in Figure 1, DFlash achieves up to a **6.1×** speedup on Qwen3-8B (Yang et al., 2025), and is nearly **2.5×** faster than the state-of-the-art EAGLE-3 across most benchmarks. We believe DFlash represents a significant step forward

in accelerating LLM inference and democratizing high-performance AI.

**Conflict of Interest Disclosure.** The authors declare no financial conflicts of interest related to this work. Research support and compute resources are acknowledged in the Acknowledgements section.

## 2. Related Work

### 2.1. Speculative Decoding

Speculative decoding accelerates LLM inference by mitigating the sequential bottleneck of autoregressive generation. Early methods (Leviathan et al., 2023) employ a smaller draft model to propose token sequences that are verified in parallel by a larger target model. Medusa (Cai et al., 2024) eliminates the external draft model by augmenting the base LLM with multiple prediction heads and using tree attention for parallel verification. The EAGLE series (Li et al., 2025c; 2024; 2025b) further improves speculative decoding by exploiting feature-level context from the frozen target model. EAGLE-1 predicts future hidden-state distributions to boost acceptance rates, EAGLE-2 introduces adaptive drafting trees, and EAGLE-3 refines training objectives to scale speedups.

Despite these advances, most existing methods rely on autoregressive drafting, which remains inherently sequential, limiting their speedups.

### 2.2. Diffusion Language Models

Diffusion large language models (dLLMs) offer an alternative to autoregressive generation by predicting masked tokens in parallel. LLaDA (Nie et al., 2025) was the first to scale dLLMs to billions of parameters, achieving performance comparable to LLaMA-3.1-8B (Grattafiori et al.,

2024). However, fully parallel diffusion models suffer from fixed-length generation and lack efficient KV cache support. Block diffusion models (Arriola et al., 2025) address these issues by denoising sequences block-by-block, blending parallelism with autoregressive structure. Building on this idea, Fast-dLLM v2 (Wu et al., 2025) and SDAR (Cheng et al., 2025) adapt pre-trained autoregressive LLMs into block-diffusion variants, enabling parallel generation while preserving generation quality on specific tasks. Nevertheless, existing dLLMs generally underperform state-of-the-art autoregressive models and often require many denoising steps, which limits their practical inference speed.

### 2.3. Diffusion-based Speculative Decoding

Recent work explores using diffusion models as drafters within speculative decoding. TiDAR (Liu et al., 2025) jointly trains diffusion and autoregressive objectives, enabling parallel "thinking" via diffusion and sequential "talking" via autoregressive decoding, though final generation quality is not yet lossless.

Other approaches repurpose autoregressive models for diffusion-style drafting. Samragh et al. (2025) observe that autoregressive LLMs implicitly encode future-token information and train a LoRA adapter to enable parallel drafting, while retaining the base model for verification.

DiffuSpec (Li et al., 2025a) and SpecDiff-2 (Sandler et al., 2025) employ large pre-trained dLLMs as speculative drafters, with inference-time search or train–test alignment to improve acceptance. However, these approaches rely on massive drafters (*e.g.*, 7B parameters), incurring substantial memory and latency overhead. While they achieve long acceptance lengths, the high drafting cost often offsets the practical speedups in real-world serving scenarios.

## 3. Preliminaries

This section formalizes the speedup mechanism of speculative decoding and clarifies the efficiency trade-offs between autoregressive and diffusion-based drafting. Our analysis highlights why diffusion drafters are uniquely positioned to achieve both low drafting latency and high acceptance rates.

### 3.1. Speculative Decoding Speedup

Speculative decoding accelerates inference of a target model $\mathcal{M}_t$ using a smaller draft model $\mathcal{M}_d$. In each decoding cycle, the draft model proposes $\gamma$ tokens, which are verified in parallel by the target model.

Following Sadhukhan et al. (2025), the average per-token latency is

$$L = \frac{T_{\text{draft}} + T_{\text{verify}}}{\tau}, \qquad (1)$$

where $T_{\text{draft}}$ is the time spent generating draft tokens, $T_{\text{verify}}$ is the cost of verification, and $\tau \in [1, \gamma + 1]$ is the expected number of accepted tokens per cycle, including the bonus token produced by the target model. Let $L_{\text{target}}$ denote the autoregressive per-token latency of $\mathcal{M}_t$; the resulting speedup is $\eta = L_{\text{target}}/L$.

This expression makes the trade-off explicit: speedup improves either by increasing the expected acceptance length $\tau$ or by reducing the drafting overhead $T_{\text{draft}}$.

### 3.2. Autoregressive *vs*. Diffusion Drafting

**Autoregressive drafters** generate tokens sequentially, incurring a drafting cost

$$T_{\text{draft}} = \gamma \cdot t_{\text{step}}, \qquad (2)$$

where $t_{\text{step}}$ is the latency of a single forward pass. Drafting costs therefore grow linearly with the speculation budget $\gamma$.

To keep latency manageable, autoregressive drafters are constrained to very shallow architectures (*e.g.*, a single transformer layer in EAGLE-3). This severely limits the draft quality: while increasing $\gamma$ increases drafting cost, acceptance length $\tau$ quickly saturates due to limited model capacity. In practice, this imbalance restricts achievable speedups.

**Diffusion drafters** generate all $\gamma$ tokens in parallel within a single forward pass, yielding

$$T_{\text{draft}} = t_{\text{parallel}}, \qquad (3)$$

where $t_{\text{parallel}}$ denotes the latency of block generation. Modern GPUs execute such parallel operations far more efficiently than multiple sequential passes, making $t_{\text{parallel}} \ll \gamma \cdot t_{\text{step}}$ for models of comparable size. For moderate block sizes, $T_{\text{draft}}$ is therefore largely insensitive to $\gamma$.

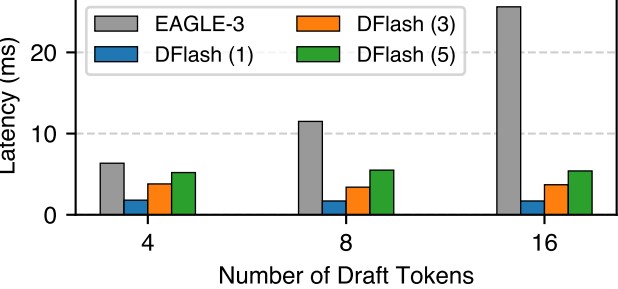

*Figure 3.* Draft cost of 1, 3, 5-layer DFlash and 1-layer EAGLE-3.

This parallelism fundamentally changes the design space. Because drafting cost no longer scales with the number of generated tokens, diffusion drafters can afford *deeper, more expressive* architectures without sacrificing latency. This increased capacity substantially improves draft quality

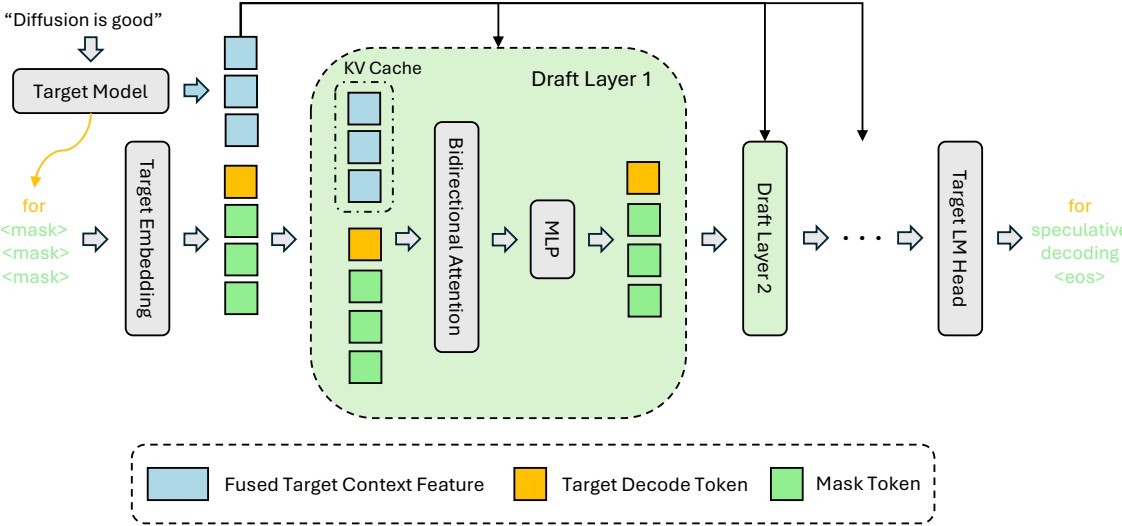

*Figure 2.* **DFlash Inference Design.** Hidden context features extracted from the target model are fused and injected into each draft layer's Key-Value cache to enable conditional speculation.

and acceptance length. Empirically, a five-layer DFlash draft model generating 16 tokens achieves both lower latency (Figure 3) and higher acceptance length than EAGLE-3 generating 8 tokens, placing DFlash on a more favorable Pareto frontier between draft quality and drafting cost.

## 4. Method

### 4.1. Inference

The system design of DFlash is illustrated in Figure 2. In this section, we explain the key design choices that allow DFlash to achieve high draft acceptance length using a very small and efficient draft model.

**Context features from the target model.** Prior work like An et al. (2025) naively applied a small diffusion model as a speculative drafter, which leads to poor acceptance length and limited speedups. To validate this, we train a five-layer block diffusion draft model *without* any conditioning from the target model and evaluate it on several math benchmarks. As the results shown in the Table 10, the resulting speedups are modest, typically around 2–3×. This limitation stems from the lack of rich contextual guidance: without access to the internal representations of the target model, the diffusion drafter must effectively predict future tokens *from scratch*.

In contrast, the hidden representations of large autoregressive target models encode substantially more information than token-level logits. These features capture long-range dependencies and task-specific semantics, and—crucially—implicitly encode information about future token predictions, as also observed by Samragh et al. (2025).

In DFlash, given an input prompt, the target model first

performs a standard prefill pass to generate the first token. During this pass, we extract hidden representations from a fixed set of layers uniformly sampled from shallow to deep. These hidden states are concatenated and passed through a lightweight projection layer to fuse cross-layer information into a compact *target context feature*, which is then used to condition the draft model.

**Conditioning via KV injection enables acceptance scaling.** Existing methods such as EAGLE-3 also leverage hidden features from the target model, but they fuse these features with the draft model's token embeddings and feed them only as inputs to the draft model. As the draft model depth increases, the information from target model becomes more and more diluted, resulting in diminishing gains in acceptance length when adding more draft layers.

DFlash adopts a fundamentally different strategy. We treat the fused target context feature as persistent contextual information and directly inject it into the Key and Value projections of *every* draft model layer. The projected features are stored in the draft model's KV cache and reused across drafting iterations. We provide more details about the KV injection mechanism in Section A.3. This design provides strong and consistent conditioning throughout the draft model, enabling acceptance length to scale effectively with the number of draft layers. We analyze this behavior in more detail in Section 5.5.2.

**Parallel diffusion drafting.** Another key contributor to DFlash's speed is its low drafting latency. Autoregressive draft models must perform multiple sequential forward passes to generate draft tokens or trees, which limits parallelism and leads to inefficient GPU utilization. In contrast, DFlash predicts the next token block using a block-level

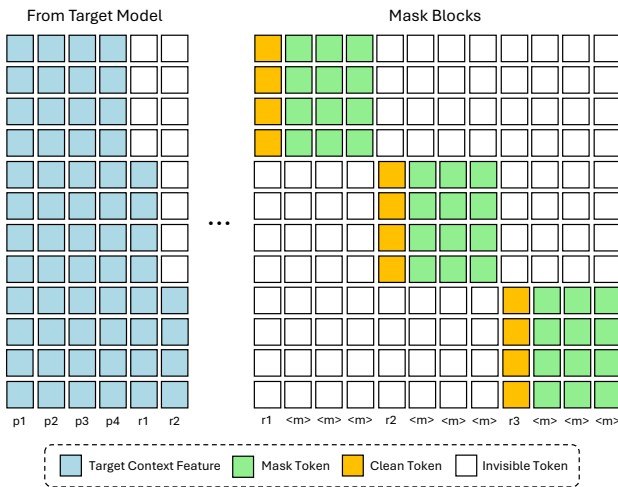

*Figure 4.* **DFlash training attention.** The target model provides context features (blue) that condition the draft model. The input consists of clean prompt tokens $p$ and clean response tokens $r$. Within each masked block, a subset of clean response tokens (yellow) is randomly sampled as anchors, while mask tokens $m$ (green) mark positions for parallel prediction. Invisible tokens (white) denote the attention mask, which enforces causal consistency and prevents inter-block information leakage during training.

diffusion process. All masked positions within a block are decoded in parallel in a single forward pass. Compared to autoregressive drafting, this block-wise parallel generation substantially reduces drafting latency and achieves significantly higher hardware utilization, even when using deeper draft models.

Overall, DFlash combines diffusion-based parallel drafting with tightly coupled conditioning from the target model, enabling high-quality drafting with substantially reduced drafting latency.

## 4.2. Training

DFlash draft models are trained to align block-level diffusion predictions with the outputs of a frozen autoregressive target model. Rather than directly adopting standard block diffusion training (Arriola et al., 2025), we introduce several key modifications that improve training efficiency, scalability, and alignment with the inference-time speculative decoding behavior.

**KV injection.** Following the inference pipeline, given a sequence consisting of a prompt and its response, we first pass the entire clean sequence through the target model to extract and fuse the hidden features for all tokens. The hidden features are then injected into the draft model as Key and Value projections, as illustrated in Figure 4.

**Random sampling of masked blocks.** In standard block diffusion training, the response is uniformly divided into blocks and random positions within each block are masked, with the model trained to denoise the masked tokens.

DFlash instead tailors block construction to the speculative decoding setting. We randomly sample *anchor tokens* from the response, use each anchor as the first position of a block, and mask the remaining positions. The draft model is trained to predict the next block_size − 1 tokens in parallel. This directly matches inference-time behavior, where the draft model always conditions on a clean token produced by the target model (*i.e.*, the bonus token from the previous verification step). Randomizing anchor positions also exposes the draft model to more diverse target context features, improving data efficiency and coverage. As shown in Table 13, this strategy substantially improves both acceptance length and speedup.

During training, all blocks are concatenated into a single sequence and processed jointly using a sparse attention mask as shown in Figure 4. Tokens attend bidirectionally within the same block and to the corresponding injected target context features, while attention across different blocks is disallowed. This design enables multiple draft blocks to be trained efficiently within a single forward and backward pass using Flex Attention (Dong et al., 2024).

**Efficient long-context training.** Training speculative draft models on long contexts is challenging for methods such as EAGLE-3 due to their costly training-time test. DFlash achieves efficient long-context training by fixing the number of masked blocks per sequence and randomly sampling anchor positions for each sequence at every epoch. This strategy provides effective data augmentation while keeping training cost bounded.

**Loss weighting for faster convergence.** In speculative decoding, *not all tokens are equal*. Errors at early positions within a draft block invalidate all subsequent tokens. This makes early predictions disproportionately important for acceptance length. We reflect this asymmetry by weighting the cross-entropy loss to emphasize earlier token positions during training.

Specifically, for a token at position $k$ within a block, we apply an exponentially decaying weight

$$w_k = \exp\left(-\frac{k-1}{\gamma}\right), \tag{4}$$

where $\gamma$ controls the decay rate. This weighting prioritizes early positions, accelerating convergence and yielding a higher acceptance length than uniform weighting (Figure 5).

**Shared embedding and LM head.** To improve training efficiency, the draft model shares the token embedding layer and language modeling head with the target model and keeps them frozen during training. Only the draft Transformer layers are updated. This design reduces the number

*Table 1.* Decoding speedup over baseline and average acceptance length ($\tau$) on Qwen3 models with thinking mode disabled and a maximum of 2048 generated tokens. Parenthesized values indicate the draft tree size for EAGLE-3 and the diffusion block size for DFlash.

| Model | Method | MATH | | | | | | CODE | | | | | | CHAT | | | |
|---|---|---|---|---|---|---|---|---|---|---|---|---|---|---|---|---|---|
| | | GSM8K | | MATH-500 | | AIME25 | | HumanEval | | MBPP | | LCB | | MT-Bench | | *Avg.* | |
| **Temperature = 0** | | Speedup | $\tau$ | Speedup | $\tau$ | Speedup | $\tau$ | Speedup | $\tau$ | Speedup | $\tau$ | Speedup | $\tau$ | Speedup | $\tau$ | Speedup | $\tau$ |
| | EAGLE-3 (16) | 1.99× | 3.30 | 1.83× | 3.08 | 1.79× | 3.05 | 1.84× | 3.05 | 1.78× | 2.95 | 1.73× | 2.91 | 1.74× | 3.02 | 1.81× | 3.05 |
| Q3-4B | EAGLE-3 (60) | 2.27× | 3.77 | 2.10× | 3.52 | 2.13× | 3.51 | 2.12× | 3.47 | 2.02× | 3.38 | 1.90× | 3.22 | 2.04× | 3.49 | 2.08× | 3.48 |
| | DFlash (16) | **5.15×** | **6.53** | **6.09×** | **7.84** | **5.68×** | **7.27** | **5.21×** | **6.64** | **4.78×** | **6.09** | **5.41×** | **7.09** | **2.85×** | **4.35** | **4.91×** | **6.54** |
| | EAGLE-3 (16) | 1.94× | 3.23 | 1.81× | 3.02 | 1.79× | 3.00 | 1.89× | 3.17 | 1.69× | 2.82 | 1.57× | 2.65 | 1.63× | 2.83 | 1.76× | 2.96 |
| Q3-8B | EAGLE-3 (60) | 2.23× | 3.71 | 2.05× | 3.49 | 2.05× | 3.44 | 2.17× | 3.65 | 1.93× | 3.25 | 1.81× | 3.03 | 1.90× | 3.26 | 2.02× | 3.40 |
| | DFlash (16) | **5.15×** | **6.54** | **6.08×** | **7.87** | **5.62×** | **7.08** | **5.14×** | **6.50** | **4.65×** | **5.95** | **5.51×** | **7.27** | **2.75×** | **4.24** | **4.86×** | **6.49** |
| **Temperature = 1** | | Speedup | $\tau$ | Speedup | $\tau$ | Speedup | $\tau$ | Speedup | $\tau$ | Speedup | $\tau$ | Speedup | $\tau$ | Speedup | $\tau$ | Speedup | $\tau$ |
| | EAGLE-3 (16) | 1.89× | 3.22 | 1.75× | 2.99 | 1.64× | 2.79 | 1.74× | 3.01 | 1.69× | 2.89 | 1.63× | 2.77 | 1.70× | 2.95 | 1.72× | 2.95 |
| Q3-4B | EAGLE-3 (60) | 2.12× | 3.68 | 1.97× | 3.44 | 1.83× | 3.20 | 1.94× | 3.39 | 1.92× | 3.33 | 1.82× | 3.14 | 1.91× | 3.36 | 1.93× | 3.36 |
| | DFlash (16) | **4.71×** | **6.00** | **5.09×** | **6.67** | **3.73×** | **4.92** | **4.74×** | **6.04** | **4.42×** | **5.66** | **4.90×** | **6.50** | **2.67×** | **4.07** | **4.24×** | **5.69** |
| | EAGLE-3 (16) | 1.87× | 3.12 | 1.73× | 2.91 | 1.63× | 2.74 | 1.75× | 3.05 | 1.64× | 2.74 | 1.56× | 2.57 | 1.58× | 2.70 | 1.68× | 2.83 |
| Q3-8B | EAGLE-3 (60) | 2.07× | 3.59 | 1.94× | 3.38 | 1.84× | 3.18 | 2.05× | 3.54 | 1.85× | 3.16 | 1.72× | 2.92 | 1.70× | 3.05 | 1.88× | 3.26 |
| | DFlash (16) | **4.67×** | **5.98** | **4.84×** | **6.40** | **3.57×** | **4.73** | **4.32×** | **5.52** | **4.04×** | **5.21** | **4.93×** | **6.69** | **2.47×** | **3.80** | **4.03×** | **5.48** |

of trainable parameters and encourages the draft model to function as a lightweight diffusion adapter tightly aligned with the target model's representation space.

## 5. Experiments

**Models and Evaluations.** We conduct experiments on LLaMA-3.1 Instruct (8B) and Qwen3 (4B, 8B, Coder-30B-A3B-Instruct) pre-trained models. We evaluate tasks in three categories: **Math:** GSM8K (Cobbe et al., 2021), MATH (Lightman et al., 2023), and AIME25 (MAA, 2025); **Code:** HumanEval (Chen, 2021), MBPP (Austin et al., 2021), and LiveCodeBench (Jain et al., 2024); **Chat:** MT-Bench (Zheng et al., 2023) and Alpaca (Taori et al., 2023). For each task, we assess the performance of the draft models using average acceptance length ($\tau$) and end-to-end decoding speedup over the autoregressive baseline. We conduct all experiments on NVIDIA H200 GPUs unless otherwise specified.

**Datasets.** To provide a diverse set of training data, we collect a mixture of around 800K samples from NVIDIA Nemotron Post-Training Dataset V2 (Nathawani et al., 2025) and CodeAlpaca (Chaudhary, 2023). Instead of directly using the original dataset, we construct our training set with the responses generated by the target model for better target alignment.

**Implementation.** For DFlash draft models, we set the number of layers to 5 (8 for Qwen3 Coder) and use a block size of 16 (10 for LLaMA 3.1). The target hidden features are extracted from 5 layers uniformly selected between the second layer and the third-to-last layer of the target model. More training details are presented in Section A.1.

**Baselines.** We compare DFlash with the vanilla autore-

gressive decoding (baseline) and state-of-the-art speculative decoding method EAGLE-3 (Li et al., 2025b). We did not include comparisons with other dLLM-based speculative decoding methods (Liu et al., 2025; Samragh et al., 2025; Li et al., 2025a; Sandler et al., 2025) due to lack of open-source implementation. For comparisons with EAGLE-3 on Qwen3 models (Section 5.1), we use the checkpoints released by AngelSlim (Tencent, 2025); for LLaMA-3.1-Instruct (Section 5.5.1), we use the official checkpoint released by EAGLE-3 team.

### 5.1. Instruct Models

In this section, we evaluate DFlash against EAGLE-3 on Qwen3 models with thinking mode disabled, using the Transformers backend. For EAGLE-3, we consider two settings: a tree size of 16, which matches DFlash with block size 16 for a fair drafting-budget comparison, and a tree size of 60, as used in the EAGLE-3 paper to maximize acceptance length with higher verification cost. In both cases, the draft steps and top-$k$ are set to 7 and 10, respectively.

As shown in Table 1, DFlash consistently outperforms EAGLE-3 across all tasks and settings. Under greedy decoding (temperature = 0), DFlash achieves an average speedup of 4.9× over the autoregressive baseline, corresponding to a 2.4× improvement over EAGLE-3 (16). Under non-greedy sampling (temperature = 1), DFlash maintains a 4.1× speedup over baseline and a 2.2× improvement over EAGLE-3. Notably, DFlash also surpasses EAGLE-3 with tree size 60, achieving higher acceptance length while incurring substantially lower verification overhead. These results demonstrate the effectiveness and efficiency of diffusion-based drafting in DFlash.

## 5.2. Reasoning Models

In this section, we evaluate DFlash for Qwen3 models with thinking mode enabled using Transformers. The draft models are trained on target-model outputs with reasoning traces.

As shown in Table 2, DFlash maintains the high acceptance length, achieving speedups of roughly $4.5\times$ and $3.9\times$ over the baseline. This efficiency gain is particularly valuable for the practical deployment of reasoning models, given their prolonged generation time.

*Table 2.* Decoding speedup over baseline and average acceptance length ($\tau$) with thinking mode enabled.

| Model | Temp. | GPQA | | MATH-500 | | AIME25 | |
|---|---|---|---|---|---|---|---|
| | | Speedup | $\tau$ | Speedup | $\tau$ | Speedup | $\tau$ |
| Q3-4B | 0 | $4.23\times$ | 5.23 | $4.59\times$ | 5.74 | $4.39\times$ | 5.54 |
| | 1 | $3.67\times$ | 4.55 | $3.93\times$ | 4.89 | $3.64\times$ | 4.68 |
| Q3-8B | 0 | $4.17\times$ | 5.17 | $4.64\times$ | 5.82 | $4.51\times$ | 5.74 |
| | 1 | $3.75\times$ | 4.65 | $4.03\times$ | 5.06 | $3.70\times$ | 4.69 |

## 5.3. Performance on Serving Frameworks

*Table 3.* Throughput (tok/s), speedup over baseline, and average acceptance length $\tau$ on SGLang (FA4 backend).

| Task | Method | Concurrency | | | | | *Avg.* $\tau$ |
|---|---|---|---|---|---|---|---|
| | | 1 | 4 | 8 | 16 | 32 | |
| **Qwen3-4B** | | | | | | | |
| Math500 | Baseline | 316 | 1145 | 2201 | 4100 | 7136 | – |
| | DFlash | 1531 | 4943 | 9066 | 14477 | 20417 | 8.01 |
| | | $4.8\times$ | $4.3\times$ | $4.1\times$ | $3.5\times$ | $2.9\times$ | |
| Human-Eval | Baseline | 312 | 1162 | 2217 | 4184 | 7143 | – |
| | DFlash | 1247 | 4147 | 6997 | 11234 | 15703 | 6.63 |
| | | $4.0\times$ | $3.6\times$ | $3.2\times$ | $2.7\times$ | $2.2\times$ | |
| **Qwen3-8B** | | | | | | | |
| Math500 | Baseline | 230 | 861 | 1666 | 3133 | 5694 | – |
| | DFlash | 1175 | 3884 | 7485 | 12268 | 16076 | 8.01 |
| | | $5.1\times$ | $4.5\times$ | $4.5\times$ | $3.9\times$ | $2.8\times$ | |
| Human-Eval | Baseline | 229 | 868 | 1649 | 3253 | 5462 | – |
| | DFlash | 955 | 3092 | 6010 | 9919 | 13116 | 6.50 |
| | | $4.2\times$ | $3.6\times$ | $3.6\times$ | $3.0\times$ | $2.4\times$ | |
| **Qwen3-Coder-30B-A3B** | | | | | | | |
| Human-Eval | Baseline | 229 | 686 | 1068 | 1681 | 2713 | – |
| | DFlash | 802 | 2078 | 3442 | 5429 | 8314 | 8.09 |
| | | $3.5\times$ | $3.0\times$ | $3.2\times$ | $3.2\times$ | $3.1\times$ | |
| LCB | Baseline | 220 | 681 | 1112 | 1733 | 2823 | – |
| | DFlash | 569 | 1621 | 2554 | 4160 | 6401 | 6.42 |
| | | $2.6\times$ | $2.4\times$ | $2.3\times$ | $2.4\times$ | $2.3\times$ | |
| MBPP | Baseline | 228 | 682 | 1057 | 1697 | 2735 | – |
| | DFlash | 720 | 2052 | 3360 | 5522 | 8538 | 7.23 |
| | | $3.2\times$ | $3.0\times$ | $3.2\times$ | $3.3\times$ | $3.1\times$ | |

In this section, we evaluate the performance of DFlash on the popular open-source inference framework SGLang using Qwen3-4B, Qwen3-8B, and Qwen3-Coder-30B-A3B-Instruct. All experiments are conducted on a single B200 GPU with the FlashAttention-4 (FA4) backend. We enable Spec-v2 scheduling overlap to maximize achievable throughput.

As shown in Table 3, DFlash consistently provides speedups across all three models over concurrency levels ranging from 1 to 32, achieving up to a $5.1\times$ speedup on Qwen3-8B. These results demonstrate the practical value of DFlash in real-world serving scenarios, where it can substantially reduce serving cost.

We report additional DFlash speedup results for more models and for vLLM (Kwon et al., 2023) in Section A.4.

## 5.4. Long Context Adaptation

In this section, we show that the base DFLash draft models trained on 4K context can adapt to longer context with minimal fine-tuning. We fine-tune the base Qwen3.5-27B draft model with 1.6K samples from LongAlign-10K (Bai et al., 2024a) for 3 epochs and test the performance on several datasets from LongBench (Bai et al., 2024b).

*Table 4.* Acceptance length of the base Qwen3.5-27B DFlash drafter (Base) and the drafter fine-tuned for long context (Long) across various context lengths on LongBench.

| Context | hotpotqa | | qasper | | gov_report | |
|---|---|---|---|---|---|---|
| | Base | Long | Base | Long | Base | Long |
| 1K | 4.91 | 4.99 | 5.27 | 5.38 | 4.53 | 4.53 |
| 2K | 4.97 | 5.06 | 5.50 | 5.67 | 4.35 | 4.38 |
| 4K | 4.91 | 5.41 | 5.17 | 5.80 | 3.93 | 4.25 |
| 8K | 4.46 | 5.76 | 4.17 | 5.62 | 3.32 | 4.04 |
| 16K | 3.61 | 6.05 | 3.57 | 6.00 | 2.67 | 3.81 |
| 32K | – | – | – | – | 2.09 | 3.56 |

As shown in Table 4, the base DFlash draft model degrades as context length grows beyond 4K, while the fine-tuned model maintains or even improves the acceptance length. This demonstrates that the extracted target features remain representative at long contexts and the draft model can learn the longer-range patterns with lightweight adaptation.

## 5.5. Ablation Study

In this section, we ablate the impact of training data and several key design choices of the DFlash draft model. Unless otherwise specified, all ablation models are trained on 100K samples randomly drawn from the full data mixture. All experiments are conducted on a single H200 GPU with greedy decoding, except those evaluated on SGLang.

### 5.5.1. TRAINING DATA

*Table 5.* Speedup over baseline and average acceptance length $\tau$ for LLaMA-3.1-8B-Instruct on SGLang (Flashinfer backend, single B200 GPU). Baseline reports absolute throughput (TPS; tokens/s). EAGLE-3 uses 7 draft steps with top-$k$=10 and either 10 or 60 draft tokens. DFlash uses block size 10.

| Method | Concurrency | | | | | Avg. $\tau$ |
|---|---|---|---|---|---|---|
| | 1 | 4 | 8 | 16 | 32 | |
| **GSM8K** | | | | | | |
| Baseline (TPS) | 249 | 923 | 1739 | 3245 | 5349 | – |
| EAGLE-3 (10) | 1.6× | 1.5× | 1.4× | 1.2× | 1.0× | 3.49 |
| EAGLE-3 (60) | 1.9× | 1.6× | 1.3× | 0.9× | 0.6× | 4.55 |
| **DFlash (10)** | **2.4×** | **2.2×** | **2.1×** | **1.8×** | **1.6×** | **4.32** |
| **HumanEval** | | | | | | |
| Baseline (TPS) | 245 | 922 | 1778 | 3336 | 5854 | – |
| EAGLE-3 (10) | 2.0× | 1.9× | 1.8× | 1.5× | 1.2× | 3.62 |
| EAGLE-3 (60) | 2.0× | 1.7× | 1.3× | 0.9× | 0.6× | 4.65 |
| **DFlash (10)** | **2.8×** | **2.6×** | **2.5×** | **2.1×** | **1.8×** | **4.91** |
| **Alpaca** | | | | | | |
| Baseline (TPS) | 245 | 906 | 1745 | 3237 | 5434 | – |
| EAGLE-3 (10) | 1.5× | 1.4× | 1.4× | 1.1× | 0.9× | 3.11 |
| EAGLE-3 (60) | 1.8× | 1.5× | 1.2× | 0.8× | 0.5× | 4.07 |
| **DFlash (10)** | **2.2×** | **2.0×** | **1.8×** | **1.5×** | **1.4×** | **3.73** |

We compare DFlash against EAGLE-3 on LLaMA-3.1-8B-Instruct. DFlash is trained on UltraChat (Ding et al., 2023) and ShareGPT, using the exactly same training data as EAGLE-3, and is evaluated against the official EAGLE-3 checkpoints. The DFlash draft model uses a block size of 10, with other configurations matching those of the DFlash Qwen3-8B draft model. All experiments are conducted using SGLang with Spec-v1 (without scheduling overlap), as Spec-v2 does not support tree-based drafting for EAGLE-3. Evaluations are performed on a single B200 GPU.

As shown in Table 5, DFlash consistently outperforms EAGLE-3 across all tasks, concurrency levels, and EAGLE-3 tree-size configurations. This performance gap holds for math, code, and chat benchmarks, demonstrating the robustness and efficiency advantages of DFlash over autoregressive tree-based speculative decoding.

### 5.5.2. NUMBER OF DRAFT LAYERS

*Table 6.* 5-layer draft model has the best average speedup. All DFlash draft models are trained with block size 16 and hidden features extracted from 5 layers of the target model.

| Setting | Math500 | | HumanEval | | MT-Bench | |
|---|---|---|---|---|---|---|
| | Speedup | $\tau$ | Speedup | $\tau$ | Speedup | $\tau$ |
| 3-L | 4.69× | 5.64 | 3.90× | 4.61 | 2.38× | 3.18 |
| 5-L | 4.71× | 5.99 | 3.96× | 4.94 | 2.35× | 3.37 |
| 8-L | 4.64× | 6.33 | 3.96× | 5.29 | 2.23× | 3.50 |

One advantage of DFlash is that acceptance length scales effectively with the depth of the draft model. However, this comes with a trade-off between drafting cost and draft quality. Deeper draft models are more expressive and achieve higher acceptance lengths, but they also incur higher drafting latency. As a result, the optimal number of layers depends on the deployment setting. As shown in Table 6, while the 8-layer draft model achieves longer acceptance lengths, the 5-layer model attains higher overall speedup due to a better balance between drafting cost and quality.

### 5.5.3. NUMBER OF TARGET HIDDEN FEATURES

*Table 7.* More hidden features from target model increases the acceptance length. All DFlash draft models use 3 draft layers and are trained with block size 16.

| Setting | Math500 | | HumanEval | | MT-Bench | |
|---|---|---|---|---|---|---|
| | Speedup | $\tau$ | Speedup | $\tau$ | Speedup | $\tau$ |
| 3-H | 4.49× | 5.38 | 3.80× | 4.47 | 2.32× | 3.07 |
| 5-H | 4.69× | 5.64 | 3.90× | 4.61 | 2.38× | 3.18 |

The number of target hidden features affects both acceptance length and end-to-end speedup. Extracting features from more target layers provides richer semantic and future-token information, improving draft quality. As shown in Table 7, conditioning on five hidden features consistently outperforms using three. However, this benefit comes at higher training cost: in offline training, the storage required to cache target hidden states increases linearly with the number of extracted features.

### 5.5.4. TRAINING-INFERENCE TIME BLOCK SIZE

*Table 8.* Ablation study of training–inference block size (BS) mismatch. All draft models use 8 layers and 5 target hidden features.

| Train BS | Test BS | Math500 | | HumanEval | | MT-Bench | |
|---|---|---|---|---|---|---|---|
| | | Speedup | $\tau$ | Speedup | $\tau$ | Speedup | $\tau$ |
| b16 | b16 | 4.64x | 6.33 | 3.96x | 5.29 | 2.23x | 3.50 |
| b16 | b8 | 3.87x | 5.09 | 3.39x | 4.44 | 2.12x | 3.18 |
| b8 | b16 | 3.78x | 5.02 | 3.24x | 4.28 | 2.09x | 3.09 |
| b8 | b8 | 3.97x | 5.21 | 3.53x | 4.61 | 2.22x | 3.29 |

Block size is a critical design choice for the DFlash draft model. An equally important question is whether a pretrained DFlash model can generalize from its training-time block size to different block sizes during inference. To study this, we train two draft models with block sizes 8 and 16 on the same data and evaluate their inference-time scaling behavior, as shown in Table 8.

When training and inference block sizes match (8→8 and 16→16), the block-size-16 model achieves substantially higher acceptance lengths on math and coding tasks. Accep-

tance histograms on Math500 reveal that the block-8 model frequently fully accepts entire blocks (35.7%), suggesting that block size 8 is often underutilized. In contrast, the block-16 model exhibits a more spread-out acceptance distribution with higher average acceptance length, indicating more effective use of larger blocks.

We further examine cross-block-size generalization at inference time and observe a clear asymmetry. A model trained with a larger block size generalizes well to smaller inference-time block sizes: using block size 8 with a model trained at block size 16 yields acceptance lengths close to those of a model trained and evaluated at block size 8. However, the reverse does not hold.

Overall, DFlash models trained with larger block sizes generalize well to smaller inference-time block sizes. This property enables dynamic block-size scheduling during inference to improve end-to-end efficiency. In practical serving scenarios, large blocks can increase verification cost under compute-bound settings (*e.g.*, large batch sizes); reducing the block size in such cases can therefore yield better overall speedup. We leave adaptive block-size scheduling to future work.

### 5.5.5. KV Injection vs. Input Fusion

This ablation studies whether target features should be injected only once at the input layer, as in EAGLE-3 style input fusion, or injected into every draft layer as KV entries. We compare these two conditioning strategies under both autoregressive drafting and block-diffusion drafting. Results are shown in Table 9.

*Table 9.* Ablation of target-feature conditioning for Qwen3-4B with 5-layer draft models and draft block size 8. Each task column reports $\tau$ / speedup.

| Variant | Injection | GSM8K | HumanEval | MT-Bench |
|---|---|---|---|---|
| *Autoregressive drafting* | | | | |
| EAGLE-3-5L | Input | 4.2 / 2.1× | 4.3 / 2.2× | 3.1 / 1.4× |
| DFlash-AR | KV | **4.8** / 2.4× | **4.6** / 2.3× | **3.4** / 1.5× |
| *Block-diffusion drafting* | | | | |
| DFlash | Input | 3.5 / 2.9× | 3.5 / 2.9× | 2.6 / 2.0× |
| DFlash | KV | **4.2 / 3.3×** | **4.0 / 3.2×** | **3.0 / 2.2×** |

The results show that KV injection is more effective than input fusion. In autoregressive drafting, DFlash-AR achieves higher acceptance length than EAGLE-3-5L on all tasks. In block-diffusion drafting, DFlash with KV injection also improves acceptance length over DFlash with input fusion on all tasks. This suggests that exposing every draft layer to target features through KV entries is more effective than injecting target features only at the input layer.

DFlash achieves acceptance length comparable to EAGLE-3-5L, but obtains much higher speedup because block dif-

fusion drafts multiple tokens in parallel. Therefore, DFlash benefits from both stronger conditioning through KV injection and faster parallel drafting.

## 6. Conclusion

In this paper, we present DFlash, a diffusion-based speculative decoding framework that rethinks the role of diffusion language models in accelerating autoregressive LLM inference. By confining diffusion models to the drafting stage, DFlash exploits their inherent parallelism while avoiding the quality degradation that has limited their standalone use. Conditioning the diffusion drafter on rich target-model context enables high acceptance rates, allowing DFlash to significantly push inference speed beyond prior speculative decoding methods.

Beyond empirical improvements, DFlash suggests a new development paradigm for diffusion LLMs. Rather than competing with autoregressive models in end-to-end generation quality, diffusion models can serve as lightweight, specialized drafters optimized for fast and accurate block prediction. This reframing permits aggressive reduction in denoising steps to maximize parallelism, while speculative verification provides a principled guarantee of output quality. We hope DFlash establishes diffusion-based drafting as a practical and scalable paradigm for speculative decoding, advancing more efficient and accessible LLM deployment.

## Acknowledgements

The authors would like to express their sincere gratitude to David Wang for leading the fast and high-quality SGLang integration for DFlash, and to Richard Gong and other members of the Modal team for their strong engineering support. Their efforts were truly instrumental in enabling the practical, production-grade deployment of DFlash.

We thank Qualcomm and Amazon for their support of this research. We also acknowledge Modal, Yotta Labs, Eigen AI, and InnoMatrix for providing the compute resources that made this work possible.

## Impact Statement

This paper presents work whose goal is to advance the efficiency of large language model inference through improved speculative decoding. The proposed method is primarily a system- and algorithm-level optimization that reduces large language model inference and serving costs, without altering model capabilities or intended use cases. We do not foresee significant new ethical risks beyond those already associated with large language models in general. Potential societal impacts are therefore consistent with existing deployments of LLMs, including both their benefits and

known limitations.

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

# A. Appendix

## A.1. Training Implementation

The draft models are optimized for 6 epochs using AdamW with a learning rate of $6 \times 10^{-4}$, a gradient clipping threshold of 1.0, and a cosine schedule with a warmup ratio of 0.04. We train on our training data mixture with a maximum sequence length of 3072 tokens (4096 for Qwen3-Coder); for each sequence, 512 anchor positions are randomly sampled. The hyperparameter $\gamma$ for the loss decay in Equation 4 is set to 7 for block size 16, 5 for block size 10, and 4 for block size 8 models.

Training can be performed either online or offline. In online training, target hidden features are computed on the fly during each training step. In offline training, target hidden features are precomputed and cached, then loaded during draft model optimization to reduce computational overhead.

## A.2. Diffusion Drafter without Target Feature

*Table 10.* A 5-layer block diffusion draft model *without* target context features. The draft model achieves only modest acceptance length and speedup.

| Temp | GSM8K Speedup / $\tau$ | Math500 Speedup / $\tau$ | AIME24 Speedup / $\tau$ | AIME25 Speedup / $\tau$ |
|---|---|---|---|---|
| 0 | 2.83 / 3.38 | 3.73 / 4.61 | 3.43 / 4.12 | 3.35 / 4.07 |
| 1 | 2.76 / 3.29 | 3.31 / 4.12 | 2.66 / 3.23 | 2.65 / 3.24 |

## A.3. KV Injection Mechanism and Memory Overhead

DFlash uses KV injection to condition the diffusion drafter on target-model features. We first concatenate hidden states from selected target layers and project them once into the draft hidden dimension:

$$\mathbf{H}_t = \text{RMSNorm}\left(W_c[\mathbf{H}^{(l_1)}; \dots; \mathbf{H}^{(l_5)}]\right).$$

The projected target features are shared by all draft layers. At layer $i$, draft tokens produce queries, while both target features and draft tokens are projected into keys and values:

$$\mathbf{Q}_i = W_i^Q \mathbf{H}_d,$$
$$\mathbf{K}_i = [W_i^K \mathbf{H}_t; W_i^K \mathbf{H}_d]_{\text{seq}},$$
$$\mathbf{V}_i = [W_i^V \mathbf{H}_t; W_i^V \mathbf{H}_d]_{\text{seq}}.$$

Thus, target features only serve as additional KV entries for the masked-block draft tokens. They bypass the draft model's $Q$ projection, output projection, self-attention update, and FFN.

The memory overhead is small. The only extra parameterized component is the shared projection $W_c \in \mathbb{R}^{D \times 5D}$. For Qwen3.5-35B-A3B with $D = 2048$ and BF16 weights, this adds

$$5 \times 2048 \times 2048 \times 2 \approx 42 \text{ MB},$$

which is negligible compared with the roughly 70 GB target model. The activation overhead is also modest: for batch size 1 and sequence length 2048, the projection input and output require about 40 MB and 8 MB, respectively. During decoding with block size 16, the temporary activation is below 400 KB.

## A.4. Results on More Models and vLLM

We further evaluate DFlash across more target models and inference frameworks. Table 11 reports SGLang results on one B200 with concurrency 8. Each entry shows acceptance length / speedup over autoregressive decoding. DFlash consistently improves over native MTP when both are available, and also scales to larger Qwen3.5, Qwen3-Coder, and GPT-OSS (Agarwal et al., 2025) models.

*Table 11.* Results across more models on SGLang. Each cell reports acceptance length / speedup.

| Model | Method | Math500 | HumanEval | MT-Bench |
|---|---|---|---|---|
| Qwen3.5-4B | MTP | 6.5 / 1.5× | 6.3 / 1.6× | 5.3 / 1.3× |
| | DFlash | 7.1 / 3.0× | 7.3 / 2.9× | 5.6 / 2.3× |
| Qwen3.5-9B | MTP | 6.7 / 1.7× | 6.6 / 1.7× | 5.3 / 1.3× |
| | DFlash | 7.3 / 3.5× | 7.9 / 3.4× | 5.5 / 2.5× |
| Qwen3.5-35B-A3B | MTP | 6.9 / 1.7× | 7.1 / 1.6× | 5.2 / 1.2× |
| | DFlash | 7.2 / 2.4× | 7.9 / 2.3× | 5.4 / 1.7× |
| Qwen3.5-27B | DFlash | 7.7 / 3.8× | 9.1 / 3.9× | 5.5 / 2.5× |
| Qwen3-Coder-Next | DFlash | 6.0 / 1.9× | 7.2 / 1.9× | 3.9 / 1.2× |
| GPT-OSS-20B | DFlash | 5.1 / 2.2× | 4.3 / 2.2× | 4.2 / 2.0× |
| GPT-OSS-120B | DFlash | 5.4 / 1.6× | 4.4 / 1.7× | 3.7 / 1.3× |

We also evaluate DFlash in vLLM on Qwen3.5-9B. Table 12 reports throughput and speedup over autoregressive decoding under different concurrency levels. DFlash achieves strong speedup at low and medium concurrency, while still maintaining throughput gains at high concurrency.

*Table 12.* vLLM results for Qwen3.5-9B. Each cell reports DFlash throughput in tok/s, with speedup over autoregressive decoding in parentheses.

| Concurrency | Math500 | HumanEval | MT-Bench |
|---|---|---|---|
| 1 | 849 (4.0×) | 969 (4.6×) | 627 (3.0×) |
| 8 | 5096 (3.2×) | 5434 (3.4×) | 3536 (2.2×) |
| 16 | 7669 (2.5×) | 8131 (2.7×) | 5297 (1.7×) |
| 32 | 9836 (1.9×) | 10258 (2.1×) | 6787 (1.3×) |

## A.5. Further Ablations

### A.5.1. LOSS DECAY

We ablate the position-dependent loss decay introduced in Section 4.2. Specifically, we compare the default setting

with exponentially decaying token weights against a variant trained with uniform token weighting within each draft block. This study isolates the effect of emphasizing early-token accuracy during training. Results in Figure 5 show that applying loss decay leads faster and better convergency.

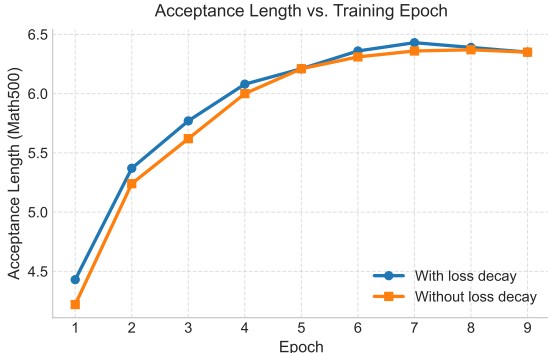

*Figure 5.* The loss decay makes training converge faster and better.

### A.5.2. RANDOM SAMPLING OF MASKED BLOCKS

*Table 13.* Randomly sample anchor tokens to construct masked blocks during training effectively augments the training data and leads to higher acceptance length and better speedup. Both draft models use three layers and extract five hidden features from the target model. The block size is 16. We use the 100K data introduced in Section 5.5 to train both models.

| Setting | Math500 | | HumanEval | | MT-Bench | |
|---|---|---|---|---|---|---|
| | Speedup | $\tau$ | Speedup | $\tau$ | Speedup | $\tau$ |
| Standard | 4.13x | 4.94 | 3.29x | 3.86 | 2.13x | 2.80 |
| **Sample** | **4.69x** | **5.64** | **3.90x** | **4.61** | **2.38x** | **3.18** |

