# OpenReview forum: "DFlash: Block Diffusion for Flash Speculative Decoding"
_ICML.cc/2026/Conference — ICML 2026 regular_

### Official Review · Reviewer_Wtpu · 2026-02-17

**Soundness:** 3
**Presentation:** 3
**Significance:** 2
**Originality:** 2
**Overall Recommendation:** 4
**Confidence:** 4

**Summary:**

While speculative decoding (SD) can accelerate inference of autoregressive LLMs, most of the existing SD methods rely on autoregressive drafting. Recently, diffusion LLMs have gained interest due to the capability of parallel decoding and there are several works that use diffusion language models as a drafter. This paper proposes DFlash, a speculative decoding method that uses a small block diffusion model for parallel drafting. DFlash generates draft tokens in a single forward pass, by conditioning the draft model on hidden features from the target model. DFlash demonstrates superior speedup over existing SD methods.

**Compliance With Llm Reviewing Policy:**

Affirmed.

**Final Justification:**

Additional experiments for weakness 3 are insightful. While I still have a concern about the novelty, I decided to increase my rating from 3 (weak reject) to 4 (weak accept) as my final rating.

**Key Questions For Authors:**

Please refer to the Weaknesses section above.

**Limitations:**

No. This paper should discuss limitations of DFlash.

**Strengths And Weaknesses:**

**Strengths**

1. The paper addresses an important research topic in speculative decoding with a dLLM drafter.
2. The paper is well-written and easy to follow.
3. Experimental results demonstrate the superior performance of DFlash.

**Weaknesses**

1. The contribution of this paper seems incremental. While it offers a valuable engineering solution, algorithmic novelty is limited. As the paper noted, parallel drafting (e.g. Medusa) and conditioning the draft model on hidden features from the target model (e.g. EAGLE) are not new.

2. This paper claims that existing methods like DiffuSpec and SpecDiff-2 are impractical due to the massive (e.g. 7B) draft models, but I believe it is an overclaim because they are using much larger target models (e.g. 32B, 70B) than this paper (e.g. 4B, 8B). If this paper wants to claim that they are impractical, it is necessary to demonstrate that DFlash shows superior speedup over DiffuSpec and SpecDiff-2 for the same target model size. Furthermore, even though there is no available implementation, I believe this paper should include more detailed discussion and comparison with these prior works using dLLM drafters to clarify the contributions of DFlash.

3. DFlash's speedup results are impressive. However, since it exploits existing ideas (parallel drafting, hidden features from the target model), I wonder what the critical factor is for this speedup. Medusa and SpecDiff also use parallel drafting, and EAGLE also conditions the draft model on hidden features from the target model. Then, is the key factor KV injection? Do you have results on DFlash (KV injection) vs DFlash (EAGLE-style condition) and DFlash (parallel drafting) vs DFlash (one-by-one drafting)?

I am open to raising my score if these points are addressed.

---

> ### Author Rebuttal · Authors · 2026-03-31
>
> We thank the reviewer for recognizing the importance of the topic, clear writing, superior results, and for the openness to raising the score.
>
> ### **Weakness 1: Novelty**
> We appreciate this concern and would like to highlight three aspects of novelty:
>
> **1. First practical diffusion drafter across diverse models and frameworks.** Unlike Medusa's independent prediction heads, DFlash models inter-token dependencies via block diffusion. It works across Qwen3/3.5 (4B–80B), LLaMA3.1, and GPT-OSS (20B–120B) on SGLang and vLLM, outperforming EAGLE-3 and native MTP. Below are SGLang concurrency=8, block size 16 results on one B200:
>
> | Model | Method | Math500 AL/Spd | Humaneval AL/Spd | MT-Bench AL/Spd |
> |--|--|:-:|:-:|:-:|
> | Qwen3.5-4B | MTP | 6.5/1.5x | 6.3/1.6x | 5.3/1.3x |
> | | DFlash | 7.1/3.0x | 7.3/2.9x | 5.6/2.3x |
> | Qwen3.5-9B | MTP | 6.7/1.7x | 6.6/1.7x | 5.3/1.3x |
> | | DFlash | 7.3/3.5x | 7.9/3.4x | 5.5/2.5x |
> | Qwen3.5-35B-A3B | MTP | 6.9/1.7x | 7.1/1.6x | 5.2/1.2x |
> | | DFlash | 7.2/2.4x | 7.9/2.3x | 5.4/1.7x |
> | Qwen3.5-27B | DFlash | 7.7/3.8x | 9.1/3.9x | 5.5/2.5x |
> | Qwen3-Coder-Next-80B-FP8 | DFlash | 6.0/1.9x | 7.2/1.9x | 3.9/1.2x |
> | GPT-OSS-20B (B8) | DFlash | 5.1/2.2x | 4.3/2.2x | 4.2/2.0x |
> | GPT-OSS-120B (B10) | DFlash | 5.4/1.6x | 4.4/1.7x | 3.7/1.3x |
>
> vLLM results for Qwen3.5-9B (DFlash tok/s, speedup vs AR):
>
> | C | Math500 | HumanEval | MT-Bench |
> |:-:|:-:|:-:|:-:|
> | 1 | 849 (4.0x) | 969 (4.6x) | 627 (3.0x) |
> | 8 | 5096 (3.2x) | 5434 (3.4x) | 3536 (2.2x) |
> | 16 | 7669 (2.5x) | 8131 (2.7x) | 5297 (1.7x) |
> | 32 | 9836 (1.9x) | 10258 (2.1x) | 6787 (1.3x) |
>
> **2. KV injection is architecturally novel.** It outperforms input fusion in acceptance length (see Weakness 3 Table) with less compute — **context tokens only pass through `k_proj`/`v_proj`, bypassing `q_proj`, `o_proj`, FFN, and self-attention**.
>
> **3. Breaking the acceptance-latency trade-off.** AR drafters' latency scales linearly with speculation budget. DFlash achieves high acceptance length with constant drafting cost — a conceptual insight validated by Section 3's analysis and experiments.
>
> ### **Weakness 2: Comparison with DiffuSpec and SpecDiff-2**
> As neither method has publicly available code, direct experimental comparison is infeasible. We compare on similar-scale targets via acceptance length, which reflects draft quality independent of system differences. DFlash with block size 16 results:
>
> | Dataset | Qwen3.5-27B | Qwen3-Coder-Next-80B-FP8 |
> |--|:-:|:-:|
> | Math500 | 7.7 | 6.1 |
> | GSM8K | 7.2 | 5.5 |
> | HumanEval | 9.1 | 7.3 |
> | MBPP | 7.3 | 5.5 |
> | MT-Bench | 5.5 | 4.1 |
>
> **vs. DiffuSpec:** Dream-7B drafter for Qwen2.5-32B, avg acceptance length 6.99 with block sizes 20–30. DFlash for Qwen3.5-27B achieves higher acceptance length with smaller block size 16 and 1.7B draft parameters.
>
> **vs. SpecDiff-2:** 7B DiffuLLAMA for LLaMA-2-70B, reports 5.04/6.57 acceptance length on Math500/HumanEval with block size 16. DFlash for Qwen3-Coder-Next-80B achieves **6.1/7.3** with 0.5B draft parameters.
>
> With a much smaller draft model, DFlash achieves higher acceptance length than both 7B-based approaches, suggesting a more favorable draft-quality–cost trade-off for accelerating large target models. At the same time, we recognize that DiffuSpec’s training-free design and SpecDiff-2’s train-time/test-time alignment are valuable contributions to diffusion drafting. These directions are complementary to DFlash, and we will add a more detailed discussion in the revised paper.
>
> ### **Weakness 3: Critical Factor for Speedup**
> To isolate each factor, we trained four draft models for Qwen3-4B under the same data, setting, and effort (block size 8):
>
> |  | GSM8K | | Humaneval | | MT-Bench | |
> |--|:-:|:-:|:-:|:-:|:-:|:-:|
> |  | AL | Spd | AL | Spd | AL | Spd |
> | EAGLE-3-5L | 4.2 | 2.1x | 4.3 | 2.2x | 3.1 | 1.4x |
> | DFlash-AR-5L (KV inj.) | 4.8 | 2.4x | 4.6 | 2.3x | 3.4 | 1.5x |
> | DFlash-inputfusion-5L | 3.5 | 2.9x | 3.5 | 2.9x | 2.6 | 2.0x |
> | DFlash-5L (KV inj.) | 4.2 | **3.3x** | 4.0 | **3.2x** | 3.0 | **2.2x** |
>
> **KV injection vs. input fusion:** Under both AR and parallel drafting, KV injection consistently achieves higher acceptance length — DFlash-AR-5L vs EAGLE-3-5L and DFlash-5L vs DFlash-inputfusion-5L. KV injection also simplifies training for AR drafters: no train-time-test with multiple forward passes as in EAGLE-3.
>
> **Parallel vs. AR drafting:** DFlash-5L has moderately lower acceptance length than DFlash-AR-5L and comparable acceptance length with EAGLE-3-5L, but achieves dramatically higher speedup (3.3x vs 2.4x and 2.1x) due to constant drafting cost. **This advantage grows with larger block sizes (e.g., 16).**
>
> The speedup stems from: (1) parallel drafting for constant cost, (2) KV injection for better quality with less compute, (3) lightweight drafter for minimal draft cost.
>
> ### **Limitations**
> We agree and will add a limitations section in the revised paper discussing compute-bound scenarios and context length generalization.

---

> > ### Author Rebuttal · Reviewer_Wtpu · 2026-04-01
> >
> > Thank you for the response. Additional experiments for weakness 3 are insightful. While I still have a concern about the novelty, I decided to increase my rating from 3 (weak reject) to 4 (weak accept) as my final rating.

---

> > > ### Author Response · Authors · 2026-04-03
> > >
> > > Thank you very much for your thoughtful review, and for raising the score and recognizing the value of the additional experiments.
> > >
> > > We also appreciate the discussion on novelty. We acknowledge that prior works such as Medusa and SpecDiff-2 explore parallel drafting, and that EAGLE-3 conditions on target hidden states. We believe DFlash’s novelty lies in the combination of block diffusion and KV injection. In our experiments, KV injection achieves higher acceptance length than EAGLE-3-style input fusion with lower compute cost, while block diffusion enables parallel drafting with constant cost. Together, they allow DFlash to achieve higher acceptance length than prior diffusion drafting methods with a much smaller drafter, and higher acceptance length than Medusa by modeling inter-token dependencies while keeping draft cost low. DFlash also demonstrates strong practical speedups for various models in real serving frameworks like vLLM and SGLang. We will clarify these novelty aspects more clearly in the revised paper. If this clarification helps address your remaining concern, we would be grateful if you would consider it in your final assessment.

---

### Official Review · Reviewer_jQ64 · 2026-03-02

**Soundness:** 3
**Presentation:** 3
**Significance:** 4
**Originality:** 3
**Overall Recommendation:** 5
**Confidence:** 4

**Summary:**

This paper introduces DFlash, a speculative decoding framework. Unlike existing methods (e.g., EAGLE-3) that rely on sequential autoregressive drafting, DFlash employs a lightweight block diffusion model for parallel draft generation. By injecting context features from the target model into the KV cache of every draft layer , DFlash achieves high acceptance rates and acceleration without compromising output quality. Results demonstrate a >6x speedup on models like Qwen3-8B.

**Compliance With Llm Reviewing Policy:**

Affirmed.

**Key Questions For Authors:**

NA

**Limitations:**

Yes

**Strengths And Weaknesses:**

Strengths
- Reduces drafting complexity from O(γ) to O(1), generating an entire block of tokens in a single forward pass.
- Innovatively injects target model features directly into the Key and Value projections of every draft layer.
- Demonstrates superior acceleration (up to 6.08x) across multiple benchmarks and shows robust performance under various concurrency levels.
- Introduces position-dependent Loss Weighting and random anchor sampling, effectively simulating the speculative environment during inference and accelerating convergence.

Weaknesses
- While diffusion models reduce forward pass counts, projecting and injecting features from 5 target model layers into all layers of the draft model increases memory footprint and computational burden. The authors should more precisely quantify the impact of this "heavy conditional injection" on inference VRAM.
- Experiments focus on sequences up to 2048 tokens. In ultra-long contexts (e.g., 32k+), are the extracted fixed-layer target features still sufficiently representative? Does the acceptance rate drop sharply due to shifts in feature distribution?

---

> ### Author Rebuttal · Authors · 2026-03-31
>
> We thank the reviewer for the careful evaluation and the recognition of DFlash's well-motivated core idea, impressive speedup results, thorough ablation studies, and clear presentation. We address the two raised concerns below.
>
> ### **Weakness 1: Memory Overhead of Conditional Injection**
>
> The memory overhead of DFlash's conditional injection is minimal. The only extra component beyond a standard draft model is a shared `fc` layer that projects the concatenated 5-layer target hidden states into the draft model's hidden dimension: `Linear(5 × D, D)`. For Qwen3.5-35B-A3B where D=2048, this adds 5 × 2048 × 2048 × 2 bytes ≈ 42 MB in weight parameters — negligible compared to the 70 GB target model.
>
> Activation memory for the projection (bsz=1, D=2048):
>
> | | Prefill (seq_len=2048) | Decode (accepted ≤ 16) |
> |--|:-:|:-:|
> | 5-layer target hidden [bsz, L, 5×D] | 5 × 2048 × 2048 × 2 = 40 MB | 5 × 16 × 2048 × 2 = 320 KB |
> | fc output [bsz, L, D] | 2048 × 2048 × 2 = 8 MB | 16 × 2048 × 2 = 64 KB |
> | Peak activation (input + output) | ~48 MB | ~384 KB |
>
> The heaviest VRAM usage occurs during prefilling, where all prefix tokens' target hidden states must be stored and projected — but at seq_len=2048, this peaks at only ~48 MB, trivial against the 70 GB target model. During decoding, the VRAM usage is minimal. All prefix tokens have been stored in draft model's KV cache, so the target hidden states correspond only to the new accepted tokens from the previous verification round, which is at most block_size (16 tokens), resulting in under 400 KB of activation memory.
>
> Once projected, the fused target hidden is simply passed into the `k_proj` and `v_proj` of each draft layer — the same KV projections that a standalone draft model or an EAGLE-3 style input fusion draft model would use anyway. However, unlike these approaches, the target context tokens in DFlash:
>
> 1. Only pass through `k_proj` and `v_proj` — they bypass `q_proj`, `o_proj`, and the entire FFN at each layer.
> 2. Do not participate in self-attention — they serve only as KV entries that the mask-block tokens query against.
>
> **This means DFlash actually reduces per-layer computation compared to an EAGLE-3 style input fusion draft model or a standalone draft model**, where the fused context tokens go through full transformer layers (Q/K/V projections, self-attention among all tokens, output projection, and FFN). In DFlash, the context tokens are **passive KV providers**, and only the small block of mask tokens (e.g., 16) undergoes the full forward pass.
>
> ### **Weakness 2: Long Context Performance**
>
> We evaluated DFlash on LongBench with Qwen3.5-27B across context lengths from 1K to 32K. The base DFlash draft model (trained on 4K context) shows acceptance length degradation as context grows beyond the training length. However, we found that **DFlash adapts to long contexts very easily**. Fine-tuning with only 1.6K samples from LongAlign-10k for 3 epochs (DFlash-Long) significantly **recovers and even improves long-context performance**. Below are the acceptance length results on LongBench:
>
> | Context | hotpotqa | | qasper | | gov_report | |
> |:-:|:-:|:-:|:-:|:-:|:-:|:-:|
> | | DFlash | DFlash-Long | DFlash | DFlash-Long | DFlash | DFlash-Long |
> | 1K | 4.91 | 4.99 | 5.27 | 5.38 | 4.53 | 4.53 |
> | 2K | 4.97 | 5.06 | 5.50 | 5.67 | 4.35 | 4.38 |
> | 4K | 4.91 | 5.41 | 5.17 | 5.80 | 3.93 | 4.25 |
> | 8K | 4.46 | 5.76 | 4.17 | 5.62 | 3.32 | 4.04 |
> | 16K | 3.61 | 6.05 | 3.57 | 6.00 | 2.67 | 3.81 |
> | 32K | — | — | — | — | 2.09 | 3.56 |
>
> *Note: only gov_report contains samples reaching 32K context length in LongBench.*
>
> At short contexts (1K), DFlash and DFlash-Long perform similarly. As context grows to 16K–32K, the base DFlash draft model degrades (e.g., hotpotqa: 4.91 → 3.61, gov_report: 4.53 → 2.09), while DFlash-Long maintains or even improves acceptance length. This demonstrates that the extracted fixed-layer target features remain representative at long contexts. The draft model simply needs lightweight context scaling training to learn the longer-range patterns.
>
> This context adaptation requires minimal effort (1.6K samples, 3 epochs), and **we will incorporate this lightweight context scaling training for all future DFlash draft models.**

---

> > ### Author Rebuttal · Reviewer_jQ64 · 2026-04-01
> >
> > I have no further questions. Good work.

---

> > > ### Author Response · Authors · 2026-04-03
> > >
> > > Thank you for the positive feedback and for confirming that your concerns have been fully addressed. We will add more details on the conditional injection design and expand the discussion of long-context generalization in the revised paper.

---

### Official Review · Reviewer_erZc · 2026-03-08

**Soundness:** 3
**Presentation:** 3
**Significance:** 4
**Originality:** 3
**Overall Recommendation:** 4
**Confidence:** 3

**Summary:**

Autoregressive LLMs are bottlenecked by sequential decoding, and while speculative decoding uses a fast draft model to mitigate this, existing methods still rely on sequential autoregressive drafting. DFlash introduces a lightweight block diffusion model that generates draft tokens in a single parallel forward pass, overcoming this limitation. By conditioning on context features from the target model, it produces high-quality drafts with improved acceptance rates. Experiments show DFlash achieves over 6× lossless speedup, outperforming the state-of-the-art EAGLE-3 by up to 2.5×.

**Compliance With Llm Reviewing Policy:**

Affirmed.

**Final Justification:**

My final rating is 4. My concern is solved.

**Key Questions For Authors:**

Could you provide more details about the block diffusion architecture? Specifically, how many denoising steps are used during inference, and how are discrete tokens mapped to and from the continuous diffusion space? Understanding these implementation details would help readers better appreciate the computational trade-offs.

**Limitations:**

See questions.

**Strengths And Weaknesses:**

The paper has several notable strengths. The core idea of using diffusion models for speculative drafting is well motivated and clearly explained. The authors provide a nice analysis of why autoregressive drafters are fundamentally limited by their linear scaling with speculation budget, while diffusion drafters can maintain constant drafting cost regardless of block size. This insight is both theoretically sound and practically important.

The experimental results are quite impressive. Achieving 4-5x speedups over baseline and consistently outperforming EAGLE-3 by a factor of 2-3x across multiple benchmarks is a significant advance. The authors evaluate on diverse tasks including math reasoning, code generation, and general chat, and they test both greedy decoding and sampling-based generation. The integration with SGLang and validation under varying concurrency levels shows that the method works in realistic serving scenarios, not just in controlled experiments.

The ablation studies are thorough and informative. The authors systematically investigate the impact of draft model depth, number of target features, and training-inference block size mismatch. The finding that models trained with larger block sizes generalize well to smaller inference-time blocks is practically useful and suggests interesting directions for adaptive scheduling.

The paper is generally well written and easy to follow. The figures and tables are clear, and the authors do a good job explaining the key concepts without getting bogged down in unnecessary technical details.

As for weaknesses, there are only a couple of minor points worth mentioning. The paper could provide more architectural details about the block diffusion model itself, such as the exact denoising schedule and how discrete tokens are handled in the continuous diffusion space. While this doesn't undermine the core contribution, additional clarity would help with reproducibility. Also, the performance on MT-Bench is noticeably lower than on math and code tasks, which is worth discussing but doesn't detract from the overall strong results.

---

> ### Author Rebuttal · Authors · 2026-03-31
>
> We thank the reviewer for the positive assessment, particularly the recognition of DFlash's well-motivated core idea, impressive experimental results, thorough ablation studies, and clear presentation. We address the two minor points below.
>
> ### **Architectural Details**
>
> DFlash is a **discrete masked diffusion model**, not a continuous diffusion model. The corruption and denoising processes operate entirely in discrete token space. Tokens are either masked or unmasked, and the model directly predicts logits over the vocabulary at each masked position. There is no continuous noise schedule or iterative denoising in embedding space. To minimize draft cost, which is a critical factor for speculative decoding speedup, we apply **one-step denoising** for the entire block of mask tokens. The draft model is forwarded only once to denoise the whole block, which is then submitted for verification. This aligns with our training setup, where we always input blocks of mask tokens starting from a clean anchor token (the bonus token from the target model) and predict the entire block in a single pass. We experimented with multi-step denoising, but the marginal acceptance length improvement did not compensate for the increased draft cost of multiple forward passes, so we use one-step denoising.
>
> ### **MT-Bench Performance**
>
> The relatively lower performance on chat benchmarks is due to two factors. First, our 800K training data mixture contains only 150K chat samples, limiting chat performance. We recently trained DFlash draft models for Qwen3.5 series with a rebalanced mixture (~30% chat data), and MT-Bench acceptance length increased significantly to ~5.5. Second, chat is inherently less structured than coding and math, making it harder for any draft model to speculate. Even the native MTP layer of Qwen3.5 models, trained jointly during pre-training, shows the same trend. Below are SGLang results (batch size 1, block size 16) comparing DFlash and MTP on Qwen3.5 models:
>
> | SGLang              | MT-Bench |         | Alpaca |         |
> |---------------------|:--------:|:-------:|:------:|:-------:|
> |                     | Acc Len  | Speedup | Acc Len | Speedup |
> | Qwen3.5-9B-MTP      |   5.3    |  1.5x   |   5.0   |  1.4x   |
> | **Qwen3.5-9B-DFlash**  | **5.5** | **3.0x** | **5.1** | **2.8x** |
> | Qwen3.5-27B-MTP     |   5.4    |  2.0x   |   5.2   |  1.5x   |
> | **Qwen3.5-27B-DFlash** | **5.5** | **3.3x** | **5.2** | **3.0x** |
>
> DFlash achieves similar acceptance length to MTP but **2x higher speedup** thanks to its parallel drafting with much lower draft cost.

---

> > ### Author Rebuttal · Reviewer_erZc · 2026-04-03
> >
> > Thanks for the experiment and response. My concerns are addressed, and I suggest that the author add the experiments to the main paper.

---

> > > ### Author Response · Authors · 2026-04-03
> > >
> > > Thank you very much for the positive feedback and for confirming that your concerns have been addressed. We also appreciate the suggestion to include these results in the main paper. In the revised version, we will add the chat and other results from our recently trained DFlash models, include more details on the rebalanced data mixture, and more clearly describe the architecture and diffusion process. If you feel these additions further strengthen the paper, we would be very grateful if you would consider this in your final assessment.

---

### Official Review · Reviewer_sNrv · 2026-03-10

**Soundness:** 3
**Presentation:** 3
**Significance:** 3
**Originality:** 3
**Overall Recommendation:** 5
**Confidence:** 5

**Summary:**

This paper proposes a new large language model inference acceleration framework named DFlash. This method uses a 5-layers block diffusion model to replace the traditional autoregressive draft model to achieve parallel draft generation. To improve the quality and acceptance rate of the diffusion model's generation, DFlash extracts and fuses the hidden layer features of the target model and injects them into the draft model's KV cache as contextual conditions. In terms of training, the paper abandons the standard random masking and instead adopts an "anchor" sampling strategy that conforms to the speculative sampling logic, and introduces exponentially decaying loss weights to emphasize the accuracy of early tokens. Experiments show that DFlash achieves significant lossless acceleration on multiple models.

**Compliance With Llm Reviewing Policy:**

Affirmed.

**Final Justification:**

Thank you for the detailed rebuttal. However, my concerns are fully resolved. I will keep my score.

**Key Questions For Authors:**

1. Regarding the KV injection mechanism: Please provide the exact mathematical formulation for the KV injection in each draft model layer. How do the draft model's own KVs and the injected Target KVs work together?
2. Regarding the decoupling of performance gains: To more clearly demonstrate the superiority of the DFlash architecture, could the authors provide a direct comparison between a "5-layer autoregressive draft model (or 5-layer EAGLE-3)" and the "5-layer DFlash model"? We need to confirm that DFlash maintains a significant advantage under strictly aligned draft model capacities. I also strongly recommend moving the results from Appendix A.2 to the main text.
3. Regarding scaling to ultra-large models: Considering the compute bound, how do the authors view DFlash's performance on ultra-large models (e.g., 100B to 235B)? Is there any quantitative analysis of the extra compute overhead (FLOPs) caused by rejection rates?
4. Regarding SGLang high-concurrency comparisons: Please supplement the direct comparison experimental data between the Qwen3 series models (especially 8B and 30B) and EAGLE-3 regarding high-concurrency throughput under the SGLang framework. Why was the EAGLE-3 baseline omitted from Table 3?

**Limitations:**

yes

**Strengths And Weaknesses:**

Strengths：
1. Clear and innovative motivation: Combining the parallel generation capability of diffusion models with the verification mechanism of speculative decoding is a highly promising research direction. This framework cleverly circumvents the poor standalone generation quality typical of diffusion models.
2. Well-designed training strategy: The anchor-based masking strategy tailored for speculative decoding and the position-weighted loss considering error accumulation are logically consistent. Ablation studies prove their effectiveness in accelerating convergence.
3. Significant single-batch acceleration: Under greedy decoding (temperature = 0), it achieves a higher single-batch speedup on the Qwen3-8B model than current state-of-the-art methods like EAGLE-3.

Weaknesses：
1. The paper lacks sufficient transparency regarding the mathematical and engineering implementation details of the draft model's internal mechanisms, which hinders reproducibility:
- Specific implementation of layer-specific projection: The paper mentions that the fused target context features are "directly inject[ed]... into the Key and Value projections of every draft model layer." However, do the layers share the same projection matrix, or does each layer have independent projection weights?
- Unclear KV Cache maintenance mechanism: If the draft model uses KV pairs projected from target hidden states, how is the KV generated during the draft model's own self-attention calculation handled? The main text lacks rigorous mathematical formulation explaining how these two coordinate (e.g., via concatenation or gating) to complete the computation.

2. DFlash defaults to a 5-layer draft model, whereas the baseline EAGLE-3 uses only a single-layer Transformer.
- Core mechanism vs. Model capacity: It is difficult for readers to intuitively judge how much of DFlash's advantage over EAGLE-3 is attributable to the "block diffusion and KV injection mechanism," and how much is simply because "the draft model is deeper and has more parameters."
- Misplacement of crucial ablations: The ablation study proving the effectiveness of KV injection (Table 8) is placed in Appendix A.2, which severely weakens the coherence of the main text's argumentation. Furthermore, this experiment only evaluates the removal of the Target Feature and lacks a strict controlled comparison against an autoregressive draft model of the same depth (e.g., 5 layers).

3. The largest model tested in the paper is the 30B-level Qwen3-Coder.
- Hidden risks of compute waste: In industrial applications, massive models of 100B or even 200B+ (e.g., 235B) have the most urgent need to solve inference bottlenecks. For such massive models, the cost of verifying large draft blocks in compute-bound scenarios is extremely high. DFlash generates relatively large blocks (e.g., 16 tokens) in parallel. If early tokens are rejected, causing massive subsequent failures, will this aggressive parallel generation result in severe redundant FLOPs overhead? The paper lacks a theoretical discussion on compute utilization.

4. High-Concurrency Evaluation：
- Misaligned core baselines: When evaluating high-concurrency scenarios using SGLang (Table 3), the paper presents the throughput of the Qwen series models but completely omits the comparison with EAGLE-3.
- Lack of head-to-head comparison for Qwen: The paper only provides a high-concurrency comparison between DFlash and EAGLE-3 on LLaMA-3.1-8B (Table 4). Given the widespread application of the Qwen series models in the industry (especially in high-concurrency settings), the absence of this critical comparison weakens the method's practical persuasiveness.

---

> ### Author Rebuttal · Authors · 2026-03-31
>
> We thank the reviewer for the thorough review, the recognition of DFlash's innovative motivation, well-designed training strategy, significant acceleration, and the detailed suggestions that help strengthen the paper.
>
> ### **Weakness 1 & Question 1: KV Injection Mechanism**
>
> A **shared** `fc` layer fuses concatenated multi-layer target hidden states into the draft hidden dimension once before the layer stack. Each draft layer uses its **own** `k_proj`/`v_proj` to project **both** target and draft representations — shared K/V projection weights within each layer. Results are concatenated along the sequence dimension. Target hidden states only pass through `k_proj`/`v_proj`, bypassing `q_proj`, `o_proj`, and FFN entirely.
> ```python
> # Shared, once before layers
> target_hidden = RMSNorm(Linear(concat(target_hiddens[selected_layers], dim=-1)))
> draft_hidden  = embed_tokens(mask_block_ids)          # [B, block_size, D]
>
> # Per draft layer i
> Q = Q_proj_i(draft_hidden)                            # [B, block_size, ...]
> K = concat([K_proj_i(target_hidden),
>             K_proj_i(draft_hidden)], dim=seq)         # [B, ctx_len+block_size, ...]
> V = concat([V_proj_i(target_hidden),
>             V_proj_i(draft_hidden)], dim=seq)
> Q, K = RoPE(Q, K)
> attn_out = Attention(Q, K, V, mask, causal=False)     # [B, block_size, ...]
> draft_hidden = draft_hidden + O_proj_i(attn_out)
> draft_hidden = draft_hidden + FFN_i(RMSNorm(draft_hidden))
> # target_hidden bypasses Q_proj, O_proj, FFN — only enters via K/V
> ```
>
> We will opensource training code for full reproducibility. For inference, we have implemented DFlash in SGLang, vLLM and MLX-LLM, which will also be opensourced.
>
> ### **Weakness 2 & Question 2: Decoupling Performance Gains**
>
> We trained three 5-layer draft models for Qwen3-4B (same data, setting, effort, block size 8):
>
> |  | GSM8K | | Humaneval | | MT-Bench | |
> |--|:-:|:-:|:-:|:-:|:-:|:-:|
> |  | AL | Spd | AL | Spd | AL | Spd |
> | EAGLE-3-5L | 4.2 | 2.1x | 4.3 | 2.2x | 3.1 | 1.4x |
> | DFlash-AR-5L (KV inj.) | 4.8 | 2.4x | 4.6 | 2.3x | 3.4 | 1.5x |
> | DFlash-5L (KV inj.) | 4.2 | 3.3x | 4.0 | 3.2x | 3.0 | 2.2x |
>
> DFlash-AR-5L vs. EAGLE-3-5L: higher acceptance length across all tasks, **showing KV injection is more effective than input fusion, independent of capacity**. DFlash-5L achieves comparable acceptance length with EAGLE-3-5L but significantly higher speedup (3.3x vs. 2.1x) via parallel drafting, breaking the acceptance-latency trade-off. For Table 8, we agree with the suggestion and will move it to the main text in the revised version.
>
> ### **Weakness 3 & Question 3: Ultra-Large Models & Compute Efficiency**
>
> We trained DFlash for larger targets. Results at Concurrency = 8 on SGLang:
>
> | Model | Math500 AL/Spd | Humaneval AL/Spd | MT-Bench AL/Spd |
> |--|:-:|:-:|:-:|
> | Qwen3.5-27B | 7.7/3.8x | 9.1/3.9x | 5.5/2.5x |
> | Qwen3-Coder-Next-80B-FP8 | 6.0/1.9x | 7.2/1.9x | 3.9/1.2x |
> | GPT-OSS-120B (B10) | 5.4/1.6x | 4.4/1.7x | 3.7/1.3x |
>
> DFlash maintains high acceptance length up to 120B. Notably, large target models are typically MoE architectures, which are less compute-bound than dense models at high concurrency, making them more friendly to DFlash's larger block sizes.
>
> Verification compute waste is a concern shared by all speculative decoding methods, but DFlash achieves better acceptance length / draft cost trade-off. Moreover, DFlash trained with B=16 adapts to smaller block sizes without retraining. We define utilization as accepted / block size. Below is the DFlash draft model for Qwen3.5-35B-A3B with block size 4, 8 and 16:
>
> | B | Math500 | HumanEval | MT-Bench | Avg AL | Util. | Wasted |
> |:-:|:-:|:-:|:-:|:-:|:-:|:-:|
> | 4 | 3.5 | 3.5 | 3.1 | 3.4 | 84% | 16% |
> | 8 | 5.5 | 5.7 | 4.4 | 5.2 | 65% | 35% |
> | 16 | 7.2 | 7.9 | 5.3 | 6.8 | 43% | 57% |
>
> At B=4, 84% of verification FLOPs produce accepted tokens — a practical knob to trade off acceptance length and compute based on the serving scenario.
>
> ### **Weakness 4 & Question 4: High-Concurrency EAGLE-3 Comparison**
>
> Due to the character limit, we provide SGLang results on the production-level 30B MoE model (Qwen3-Coder-30B-A3B) here. EAGLE-3: depth 8, top_k=10, tokens 16/60. DFlash: block size 16.
>
> | C | Method | Humaneval AL/Spd | MBPP AL/Spd |
> |:-:|--|:-:|:-:|
> | 1 | EAGLE-3 (16) | 6.2/1.9x | 6.5/2.0x |
> | 1 | EAGLE-3 (60) | 6.9/1.9x | 7.2/2.3x |
> | 1 | **DFlash (16)** | **8.1/3.5x** | **7.2/3.2x** |
> | 8 | EAGLE-3 (16) | 6.2/1.8x | 6.5/2.0x |
> | 8 | EAGLE-3 (60) | 6.8/1.5x | 7.2/1.9x |
> | 8 | **DFlash (16)** | **8.1/3.2x** | **7.2/3.2x** |
> | 16 | EAGLE-3 (16) | 6.2/1.5x | 6.5/1.9x |
> | 16 | EAGLE-3 (60) | 6.8/1.2x | 7.2/1.6x |
> | 16 | **DFlash (16)** | **8.1/3.2x** | **7.2/3.3x** |
> | 32 | EAGLE-3 (16) | 6.2/1.2x | 6.5/1.5x |
> | 32 | EAGLE-3 (60) | 6.8/1.0x | 7.2/1.3x |
> | 32 | **DFlash (16)** | **8.1/3.1x** | **7.2/3.1x** |
>
> DFlash maintains ~3x speedup across all concurrency levels, while EAGLE-3 degrades significantly. We will add more Qwen model comparisons on SGLang in the updated paper.

---

> > ### Author Rebuttal · Reviewer_sNrv · 2026-04-03
> >
> > No more questions.

---

### Decision · Program_Chairs · 2026-04-30

**Decision:**

Accept (regular)

**Comment:**

This paper introduces DFlash, a speculative decoding framework that replaces traditional sequential autoregressive drafting with a lightweight block diffusion model. By leveraging parallel generation, DFlash reduces the drafting complexity from $O(\gamma)$ to $O(1)$, effectively breaking the latency bottleneck inherent in current state-of-the-art methods like EAGLE. A key technical contribution is the KV injection mechanism, which fuses hidden features from the target model directly into the draft model’s layers to improve acceptance rates without significant computational overhead.

The reviewers are in consensus that the paper is technically solid and offers a significant practical advancement for LLM inference. The authors have committed to including the additional ablation studies and long-context results in the final version. Given the high impact on inference efficiency and the thoroughness of the response, I recommend this paper for Acceptance.